# The Association between Term Chorioamnionitis during Labor and Long-Term Infectious Morbidity of the Offspring

**DOI:** 10.3390/jcm13030814

**Published:** 2024-01-31

**Authors:** Noa Efrat Davidi, Gil Gutvirtz, Eyal Sheiner

**Affiliations:** 1Faculty of Health Sciences, Joyce and Irving Goldman Medical School, Ben Gurion University of the Negev, Beer-Sheva 84105, Israel; noaefratdavidi@gmail.com (N.E.D.); giltzik@gmail.com (G.G.); 2Department of Obstetrics and Gynecology, Soroka University Medical Center, Beer-Sheva 84101, Israel

**Keywords:** term pregnancy, chorioamnionitis, labor, long-term, follow-up, infectious morbidity

## Abstract

**Background**: Chorioamnionitis during labor exposes the fetus to an intrauterine state that may alter the future immune response and may expose the offspring to future susceptibility to infectious disease. We evaluated the long-term pediatric infectious morbidity of children born at term to mothers who have chorioamnionitis during labor. **Methods:** This was a population-based cohort analysis including only term singleton deliveries at a regional tertiary hospital between the years 1991 and 2021. Offspring to mothers with and without a diagnosis of chorioamnionitis during labor were compared. Offspring hospitalizations up to the age of 18 years involving infectious morbidity were evaluated using the Kaplan–Meier survival curve and a Cox regression model to control possible confounders. **Results:** A total of 331,598 deliveries were included, 988 (0.3%) of which were of mothers diagnosed with chorioamnionitis during labor. All infectious morbidity rates included in the analysis were comparable between groups. The Kaplan–Meier survival curves were similar for both groups (log-rank = 0.881) and the multivariable analysis ascertained that chorioamnionitis during labor was not a risk factor for offspring’s long-term infectious morbidity (HR 0.929, 95%CI 0.818–1.054, *p* = 0.254). **Conclusions:** In our cohort, term chorioamnionitis during labor was not associated with a higher risk of pediatric hospitalization due to infections. The infectious/inflammatory state during labor did not expose nor increase the susceptibility of the term offspring to future infectious morbidity.

## 1. Introduction

Chorioamnionitis (intra-amniotic infection (IAI)) refers to a state of infection or inflammation of intrauterine elements including the fetus, fetal membranes, amniotic fluid, umbilical cord, and/or the placenta. Due to the heterogeneity of the disorder, an expert panel from the NICHD recommended the use of the term “triple I”, indicating a state of intrauterine inflammation, infection, or both [1]. Although the latter expert panel defined “triple I” using strict diagnostic criteria to minimize interobserver heterogeneity, it is still often diagnosed clinically, when indicated by fever, maternal or fetal tachycardia, uterine tenderness, and foul-smelling vaginal discharge [2]. The incidence of chorioamnionitis varies between studies mainly due to inconsistent diagnostic criteria [1,3], but is far more prevalent in preterm deliveries, with incidence ranging from 40% to 70% in pre-term deliveries and only 1% to 4% among term deliveries [2]. As such, chorioamnionitis is thought to be a leading cause of preterm birth [4].

The most common pathway to IAI is the ascent of pathogens from the lower genital tract to the amniotic cavity. Others are the hematogenic pathway because of maternal bacteremia or the iatrogenic pathway due to an invasive procedure like amniocentesis or fetoscopy [5,6].

In most cases, the infection is polymicrobial. The most frequent pathogens are genital Mycoplasmas: Ureaplasma and Mycoplasma species [7]. The maternal and fetal immune response may lead to the release of prostaglandins, membrane injury, ripening of the cervix, and eventually preterm labor [8]. Multiple risk factors for chorioamnionitis have been established. Among them are prolonged membrane rupture and labor, multiple digital vaginal examinations, nulliparity, meconium-stained amniotic fluid, colonization of genital tract pathogens (group B Streptococcus, Bacterial Vaginosis), epidural anesthesia, and more [2]. Adverse maternal outcomes include increased risk for cesarean delivery, uterine atony, postpartum bleeding, need for blood transfusion, and sepsis [1,9]. Detrimental neonatal outcomes are preterm labor [10] exposing the offspring to the complications of prematurity, asphyxia, early-onset neonatal sepsis, respiratory distress syndrome (RDS), pneumonia, meningitis, intraventricular hemorrhage, cerebral white matter damage, and perinatal death [9,11].

Whether the surviving offspring of mothers who have chorioamnionitis suffer long-term consequences is unclear. Some studies found chorioamnionitis to be associated with neurological disorders and specifically cerebral palsy [11,12,13]. Moreover, an association between chorioamnionitis and an increased risk of respiratory morbidities such as asthma and recurrent wheezing was also reported [14].

Although notable achievements have been accomplished in the reduction in the incidence of childhood infectious morbidities, they are still considered the leading cause of death for children under the age of five according to the WHO 2019 report [15]. To the best of our knowledge, there are no studies to date that investigate long-term infectious morbidity in the offspring of mothers who were diagnosed with chorioamnionitis during pregnancy.

Hence, in this large population-based study, we sought to further investigate the long-term (up to the age of 18) infectious outcomes of children to mothers who had chorioamnionitis during labor, compared with children who were born to mothers without such a diagnosis.

## 2. Materials and Methods

This is a retrospective cohort study including all singleton term deliveries that occurred between 1991 and 2021 at Soroka University Medical Center (SUMC), a tertiary hospital in the Negev region (southern Israel). Hence, the data is comprised of a nonselective population. Although IAI is notably more common in preterm deliveries, these are also associated with much higher rates of infant mortality and morbidity, including infectious morbidities [16]. Therefore, we decided not to include preterm deliveries in our study. Using this strategy allowed for a better, cleaner clinical data set that also excluded possible infectious etiologies leading to premature labor. Multiple pregnancies and offspring with congenital abnormalities were also excluded from the study, as these children are also at an increased risk for future complications and childhood morbidities unrelated to chorioamnionitis during labor. Perinatal death cases were excluded from the long-term analysis.

Maternal data included maternal age and parity as continuous variables, and obesity (defined as BMI ≥ 30 kg/m^2^), smoking status (yes/no), use of fertility treatments (yes/no), and other medical comorbidities including diabetes mellitus and hypertensive disorders as binary variables (yes/no).

We compared the incidence of childhood infectious morbidity (up to the age of 18) in offspring born to mothers who did and did not have chorioamnionitis during term delivery.

The exposure variable, chorioamnionitis, was diagnosed and documented by senior attending obstetricians during labor. In SUMC, the diagnosis is made primarily by clinical criteria including fever (>38.0 °C) and/or maternal tachycardia (heart rate > 100), and/or fetal tachycardia (heart rate > 160), and/or uterine tenderness, and/or foul-smelling amniotic fluid. To establish the diagnosis, parturients must present with at least two of the abovementioned clinical criteria; however, not all criteria must be fulfilled.

Adverse perinatal sequels were assessed, including gestational age at birth, birthweight, rates of meconium-stained amniotic fluid, prolonged 2nd stage of labor (defined as >3 h in the 2nd stage of labor at full dilation), cesarean delivery rates, low 5 min Apgar scores (defined as Apgar score < 7), small for gestational age (SGA) (defined as birthweight < 5th percentile for gestational age and gender), low birthweight (defined as birthweight < 2500 g), postpartum hemorrhage rates and perinatal mortality rates (including stillbirth, intrapartum fetal death, and neonatal death).

Finally, long-term infectious-related hospitalizations of the offspring up to the age of 18 years were evaluated. The different infectious morbidities assessed in this study were predefined by ICD-9 codes for pediatric infectious morbidity and are presented in the Appendix A).

Follow-up time was represented by the time elapsed before an event (first hospitalization with any of the infectious diagnoses or until censored). Censored cases are cases of child death (during hospitalization unrelated to infectious diseases), when a child reaches age 18 years (calculated based on the date of birth), or at the end of the study period. Only the first hospitalization with an infectious diagnosis for each child was included in the analysis.

Data were assembled from two databases that were cross-linked using the mother and child IDs. The first database is the computerized perinatal database of the obstetrics and gynecology department, containing demographic information and maternal and neonatal information recorded at hospital admission and immediately following delivery by an obstetrician. The second is the computerized pediatric hospitalization database, which includes ICD-9 codes for all medical diagnoses made during child hospitalizations in the SUMC pediatric departments. All medical files, maternal, perinatal, and offspring hospitalization records are routinely revised by qualified medical secretaries before archiving.

Statistical analysis was performed using the SPSS package 29th ed. (SPSS, Chicago, IL, USA). The general association between categorical data was assessed via chi-square. For comparison differences in continuous variables, the Student t-test was used. The Kaplan–Meier survival curve was used to compare cumulative hospitalization rates due to infectious morbidities according to maternal chorioamnionitis status during labor. The variance between the curves was assessed using the log-rank test. A Cox proportional hazards model was used to investigate the association between chorioamnionitis and offspring infectious morbidity. The model was adjusted for possible confounders including maternal age at delivery, gestational age (across term), mode of delivery, maternal diabetes mellitus, and hypertensive disorders. All analyses were two-sided, and a *p*-value of ≤0.05 was considered statistically significant.

## 3. Results

During the study period, 331,598 term singleton deliveries in SUMC met the inclusion criteria, of which 988 (0.3%) were of mothers that were diagnosed with chorioamnionitis during labor. The maternal characteristics of the study population are described in Table 1 presented below. Chorioamnionitis has been recorded more often in older women, who were more often nulliparous. These women were also more likely to have undergone fertility treatments. Obesity rates were also higher in the chorioamnionitis group, as well as maternal smoking rates. The prevalence of maternal hypertensive diseases and maternal diabetes mellitus was also significantly higher in the chorioamnionitis group.

Table 2 presents pregnancy outcomes for deliveries of women who were diagnosed with chorioamnionitis and those who were not. Deliveries complicated with chorioamnionitis were more often induced into labor and had higher rates of meconium-stained amniotic fluid during labor. They also had a prolonged 2nd stage of labor, which may have also led to higher rates of cesarean deliveries. Postpartum hemorrhage was also more common in this group. Women with chorioamnionitis delivered at a statistically higher mean gestational age. However, mean birthweight was lower in the offspring of affected mothers, and they were more likely to be small for their gestational age (SGA). Notably, although gestational age and mean birthweight were found to have a statistically significant difference between the two study groups, this difference is clinically irrelevant as both outcomes reside well within the full term of gestational age with an appropriate birthweight. The chorioamnionitis group of children had higher rates of low Apgar score at 5 min and the perinatal mortality rate was significantly higher in this group.

Table 3 illustrates the long-term infectious morbidities of children (up to the age of 18 years) born to mothers who did and did not have chorioamnionitis and the total infectious-related hospitalization rate. All infectious morbidities evaluated in this study and the total infectious-related hospitalization rate were comparable between the groups.

The Kaplan–Meier survival curve presented in the Figure 1 demonstrates a comparable cumulative incidence of infectious-related hospitalizations during the study period between children born to mothers who did have chorioamnionitis during labor and those who did not.

Table 4 presents the multivariable analysis for the association between chorioamnionitis during labor and offspring long-term infectious morbidity. The model was adjusted for possible confounding and clinically important variables including maternal age, gestational age, maternal diabetes, hypertensive diseases, and mode of delivery. The model revealed that chorioamnionitis during term labor was not associated with long-term infectious morbidity of the offspring.

## 4. Discussion

Chorioamnionitis during pregnancy was previously associated with significant maternal, perinatal, and long-term adverse outcomes for mothers and children. The long-term outcomes of children born to mothers who were diagnosed with chorioamnionitis include a higher risk of lung disease, asthma, cerebral palsy, and other neurodevelopmental disabilities [13,14]. It has been postulated that chorioamnionitis can potentially influence the development and maturation of the neonatal immune system and that in utero exposure to an inflammatory state such as chorioamnionitis, even in the absence of infection, may “prime” the developing immune system, resulting in a more activated immunophenotype, potentially increasing the susceptibility of the infant to later childhood diseases, altering their response to vaccination, or contributing to the development of immunopathological disorders [17]. It has been shown that the offspring of mothers who have chorioamnionitis during labor are at higher risk for short-term perinatal infectious morbidity like sepsis and meningitis [18,19,20]. However, to the best of our knowledge, we found no previous studies evaluating whether these children also suffer long-term childhood infectious morbidity. Hence, this study aimed to evaluate a possible long-term infectious outcome.

Interestingly, a very recent article by Zamstein et al. [21] investigated the long-term infectious morbidity of infants exposed to intrapartum maternal fever (but not chorioamnionitis per se). They describe higher rates of long-term pediatric infectious morbidity in children that were exposed to intrapartum maternal fever. In our cohort, we found no significant association between chorioamnionitis during labor and long-term infectious morbidity of the offspring. None of the pediatric infectious morbidities investigated in our cohort were found to be more prevalent in affected children, and the risk for infectious-related hospitalization was comparable to non-exposed children. One possible explanation for the discrepancy between studies may reside in the fact that Zamstein et al. [21] included all cases of intrapartum fever, regardless of the cause of maternal fever, which may impact neonates’ health differently. They also included preterm deliveries in their cohort, and prematurity was twice as high in the maternal fever group compared to the unexposed group. Additionally, when examining the multivariable regression model used in their study, gestational age seems to have a protective effect for long-term infectious morbidity, so our choice to use only term-delivered children may provide some explanation for our results.

The management of suspected chorioamnionitis during labor dictates admission of broad-spectrum antibiotic agents for the mother [22]. Antibiotic treatment may impact the newborn microbiome and, in turn, alter the host organ functions including the immune system [23]. Nevertheless, we did not find changes in long-term infant infection hospitalization rates between groups, providing the first reassuring evidence regarding the clinical impact of chorioamnionitis on future infant susceptibility to significant infectious diseases. However, previous studies have indicated that antibiotic use in the first year of life can increase the chance of childhood overweight and diabetes mellites [24,25]. Moreover, epidemiological studies have shown the association between maternal exposure to antibiotics in the peripartum period and the increased risk of diseases like asthma, atopic dermatitis, food allergies, and IBD [26,27,28]. Those morbidities involve an overactive immune response to normally harmless substances, leading to inflammation and tissue damage.

Furthermore, some factors have a positive effect on the functioning and development of the immune system of infants: breastfeeding, vaccinations, probiotics, and nutrition [29]. We could assume that, considering those other factors, the changes that happen in the offspring might be temporary and their immune systems recover over time.

When chorioamnionitis occurs, microbial products are found in the amniotic fluid. Therefore, direct colonization of bacteria, bacterial by-products, and other bacterial matters can occur in the fetus [5]. This could be an explanation for the short-term risk of sepsis and infection in the neonatal period. Studies show that exposure to histologic chorioamnionitis alters the neonatal immune transcriptosome, with activation of innate immune pathways [30]. Berwick et al. [31] have shown that immune gene expression was decreased in chorioamnionitis-exposed monocytes.

Those studies relate to a short time of immunological influences and mostly discuss the innate immune system. As a newborn grows up, several molecular changes occur within the innate immune system, contributing to its improved ability to respond to infections and pathogens. For example, increased expression of pattern recognition receptors (PRRs) such as toll-like receptors (TLRs) play a crucial role in recognizing and responding to pathogens. The elevated production of cytokines, including interleukins and interferons, which serve as signaling molecules and regulate the immune response [32], increases the number of dendritic cells which serve as the gatekeepers between the innate and adaptive immune response [33]. In newborns and young children, the innate immune system is the dominant form of defense against infections. As children grow and develop, the adaptive immune system, which includes B and T cells, becomes increasingly important in providing immunity to specific pathogens [34]. These processes potentially account for the lack of substantial influence on infectious disease rates among offspring exposed to chorioamnionitis at term birth.

Moreover, the link between chorioamnionitis at term and long-term infectious diseases may be skewed by confounding factors such as maternal health, immunological status, and overall health behaviors. Antenatal care during pregnancy, inclusive of access to preventative and therapeutic measures, can affect the risk of infectious disease. Socioeconomic status is recognized as a significant determinant affecting infectious morbidity and mortality [35], in addition to healthcare access and preventative measures. Genetic factors also contribute to the risk of infectious disease development [36].

Finally, as noted before, chorioamnionitis is much more common among preterm deliveries and is probably a major contributor to the pathologic premature process of labor. Choosing only term infants may suggest that the inflammatory process may have only developed during labor and did not reside in utero for a long time, so its effect on the fetus may be less pronounced than in preterm deliveries.

The major strength of our study is its considerable size and non-selective population base. SUMC is the sole tertiary hospital treating the population of southern Israel, which historically has a positive net migration rate. SUMC performs most birth procedures, follow-ups, and pediatric care in the south of Israel. Most babies born in SUMC are likely to return for medical treatment and follow-up. This fact allows us a sequential patient follow-up. All details were documented by an expert medical staff of obstetricians and pediatricians, revised by trained medical secretaries for accuracy, and digitally archived, which allows ample analysis of maternal, obstetric, and pediatric data. This poses a unique opportunity for us, as we can extract data from hospitalizations of offspring in pediatric wards and match it with maternal obstetric data by using their given ID numbers. Having a large database also allows for the detection and inclusion of less common diagnoses that smaller studies do not have access to in order to make an accurate evaluation regarding their true prevalence.

However, our study has limitations. It focuses on diagnoses of infections during a child’s hospitalization, yet it is recognized that the majority of childhood infections do not necessitate hospital admission. As such, these cases would not be represented in our data. This focus could potentially lead to an underestimation of the true impact of chorioamnionitis exposure on infection rates. Therefore, this aspect should be considered when interpreting our findings.

For the reasons described above, we decided to investigate only term singleton deliveries without any known congenital malformation, which enabled us to focus on otherwise healthy term born infants, cancelling out the impact of multiple fetuses and prematurity in our cohort. However, this decision also led to an inevitable “black-hole” regarding our knowledge in cases of chorioamnionitis in the setting of multiple pregnancies or preterm deliveries, with the latter constituting the majority of chorioamnionitis cases. Further research is imperative in these unexplored settings.

Another limitation to note is the relatively high mortality rate (5.4%) among infants diagnosed with chorioamnionitis. This could suggest a possible selection bias toward more severe cases within this group. However, this study focused on offspring long-term infectious outcomes and despite this bias, it is reassuring that even among this severely affected population, we did not observe differences in future rates of infectious morbidity.

In addition, our study does not include childhood environmental exposure, lifestyle factors, and socioeconomic status, all of which may play an important role in many health aspects for developing children. These factors are virtually impossible to control in a retrospective study, but since the study population resides in the same area, it is possible to presume some similar exposures, which hopefully have only minimal effects on the results.

It is also important to note that the diagnosis of chorioamnionitis is heterogenic, and some variations may have also existed among physicians who registered the diagnosis during the ample study period. However, SUMC holds a protocol for the diagnosis and management of chorioamnionitis, which includes some specific criteria, but also allows a diagnosis based on clinical suspicion, which is based on the obstetrician’s subjective impression. We cannot assess which of these criteria were implemented in each exposed case.

We acknowledge that we were unable to include the placental pathology report that would have confirmed the diagnosis in these deliveries, or further investigate the association between the histological grading of chorioamnionitis and the long-term infectious outcome of the offspring. Further studies are needed to evaluate this topic.

Finally, in this study, we were unable to include documentation of antibiotic regimens given to mothers diagnosed with chorioamnionitis during labor, but we can safely assume that since it is the common practice to administer antibiotics in these settings, all cases were treated prenatally. Importantly, antibiotic treatment during labor may alter the disease course and potentially protect the fetus from acquiring bacterial infections. However, in utero exposure to antibiotics may also increase the child’s susceptibility to childhood infections [37]. In the case of chorioamnionitis during labor, and specifically in term fetuses, we speculate that the very brief exposure to antibiotics produces a negligible effect on the long-term infectious morbidity of these children.

## 5. Conclusions

Our study contributes to the understanding of the consequences of chorioamnionitis during term labor on offspring long-term infectious health, as we found no association with later infectious morbidity. The unique setting of SUMC described earlier allows for a long-term follow-up on children born in the hospital and later treated in our pediatric department. Considering the limitations of the study, it provides some reassurance that the temporary in utero exposure to maternal chorioamnionitis did not affect the long-term infectious health of the children in our cohort. We hope that these findings might alleviate some of the stress and concerns of parents when facing this stressful scenario.

## Figures and Tables

**Figure 1 jcm-13-00814-f001:**
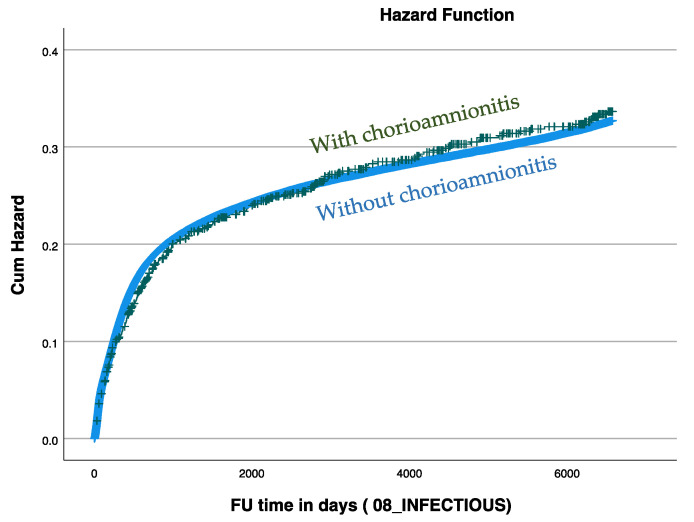
Kaplan–Meier survival curves demonstrating the cumulative incidence of infectious-related hospitalizations between study groups (log-rank = 0.881).

**Table 1 jcm-13-00814-t001:** Maternal characteristics of the study population.

Characteristics	With Chorioamnionitis [*n* = 988]	Without Chorioamnionitis [*n* = 330,610]	*p*-Value
Maternal age (mean, ±SD)	29.0 ± 6.1	28.2 ± 5.7	0.003
Nulliparity, *n* (%)	423 (42.8%)	72,478 (23.7%)	<0.001
Obesity, *n* (%)	19 (1.9%)	3760 (1.1%)	0.02
Smoking, *n* (%)	13 (1.3%)	2274 (0.7%)	0.017
Fertility treatments, *n* (%)	30 (3.0%)	5195 (1.6%)	<0.001
Hypertensive disorders, *n* (%)	94 (9.5%)	13,612 (4.1%)	<0.001
Diabetes mellitus, *n* (%)	93 (9.4%)	15,270 (4.6%)	<0.001

**Table 2 jcm-13-00814-t002:** Pregnancy outcomes for deliveries of women with and without chorioamnionitis.

Pregnancy Outcome	With Chorioamnionitis [*n* = 988]	Without Chorioamnionitis [*n* = 330,610]	*p*-Value
Gestational age at birth (weeks, ±SD)	39.5 ± 1.3	39.3 ± 1.2	<0.001
Birthweight (g., ±SD)	3185 ± 527	3263 ± 443	<0.001
Induced labor, *n* (%)	436 (44.1%)	70,788 (21.4)	<0.001
Meconium-stained amniotic fluid, *n* (%)	373 (37.8%)	40,982 (12.4%)	<0.001
Prolonged 2nd stage of labor, *n* (%)	47 (4.8%)	4169 (1.3%)	<0.001
Cesarean delivery, *n* (%)	617 (62.4%)	41,746 (12.6%)	<0.001
Low 5 min Apgar score ^a^, *n* (%)	31 (3.3%)	1238 (0.4%)	<0.001
SGA ^b^, *n* (%)	117 (11.8%)	14,821 (4.5%)	<0.001
Low birthweight ^c^, *n* (%)	98 (9.9%)	11,434 (3.5%)	<0.001
Postpartum hemorrhage, *n* (%)	13 (1.3%)	1937 (0.6%)	0.003
Mortality, *n* (%)	53 (5.4%)	860 (0.3%)	<0.001

^a^ Apgar < 7 at 5 min. ^b^ SGA, small for gestational age, <5th percentile for gestational age and gender. ^c^ Birthweight < 2500 g.

**Table 3 jcm-13-00814-t003:** Long-term infectious morbidities in children (up to the age of 18 years) born to mothers who did and did not have chorioamnionitis.

Infectious Morbidity	With Chorioamnionitis [*n* = 935]	Without Chorioamnionitis [*n* = 329,750]	*p *-Value
Bacterial, *n* (%)	18 (1.9%)	4589 (1.4%)	0.165
Viral, *n* (%)	34 (3.6%)	10,408 (3.2%)	0.402
Respiratory, *n* (%)	135 (14.4%)	44,582 (13.5%)	0.412
Central nervous system (CNS), *n* (%)	7 (0.7%)	1678 (0.5%)	0.304
Ear-nose-throat (ENT), *n* (%)	63 (6.7%)	20,947 (6.4%)	0.629
Gastrointestinal (GI), *n* (%)	31 (3.3%)	9650 (2.9%)	0.481
Ophthalmic, *n* (%)	12 (1.3%)	4039 (1.2%)	0.871
Dermatologic, *n* (%)	23 (2.5%)	7583 (2.3%)	0.744
Total infectious hospitalizations, *n* (%)	242 (25.9%)	78,409 (23.8%)	0.131

**Table 4 jcm-13-00814-t004:** Cox regression model for the association between chorioamnionitis in term pregnancy and offspring long-term infectious morbidity.

Variables	Adjusted Hazards Ratio (aHR)	95%CI		*p*-Value
		Min	Max	
Chorioamnionitis	0.929	0.818	1.054	0.254
Mother’s age at birth	0.988	0.987	0.990	<0.001
Gestational age in weeks	0.940	0.935	0.945	<0.001
Cesarean section	1.091	1.068	1.114	<0.001
Hypertensive disorders ^a^	1.037	1.002	1.073	0.036
Any diabetes ^b^	1.083	1.048	1.119	<0.001

^a^ Including pregestational, gestational hypertension, and pre-eclampsia. ^b^ Including pregestational and gestational diabetes.

## Data Availability

The original contributions presented in the study are included in the article/Appendix A, further inquiries can be directed to the corresponding author.

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
