# Peer review of "The Association between Term Chorioamnionitis during Labor and Long-Term Infectious Morbidity of the Offspring"

_jcm, 2024, doi:10.3390/jcm13030814_

Round 1

Reviewer 1 Report

Comments and Suggestions for Authors

Dear authors,

I was pleased to read the article entitled: The association between term chorioamnionitis during labor and long-term infectious morbidity of the offspring.

The introduction introduces the topic of the article in a fair way, the results were clearly presented. I would suggest adding in the Materials and Methods section what mother's data were analyzed, or just those shown in Table 1? 

I have no comments on the Discussion and Results section.

Author Response

Dear authors,

I was pleased to read the article entitled: The association between term chorioamnionitis during labor and long-term infectious morbidity of the offspring.

The introduction introduces the topic of the article in a fair way, the results were clearly presented. I would suggest adding in the Materials and Methods section what mother's data were analyzed, or just those shown in Table 1? 

Answer: We thank the reviewer for the kind remarks. We added the maternal data that was collected and analyzed in the article to the Materials and Methods section as follows: “Maternal data included maternal age and parity as continuous variables, obesity (defined as BMI≥30 kg/m2), smoking status (yes/no), use of fertility treatments (yes/no) and other medical comorbidities including diabetes mellitus and hypertensive disorders as binary variable (yes/no).” (Page 2, lines 81-84).

I have no comments on the Discussion and Results section

We thank the reviewer for the contribution to the study.

Reviewer 2 Report

Comments and Suggestions for Authors

Thankyou to the authors for an interesting paper.  The paper is well written and a good read.

In this manuscript the authors investigate the hypothesis that term chorioamnionitis would be associated with a long term and higher infectious morbidity of the affected infants.

This is a retrospective study and had impressive numbers of patients/ data with an n of 330,610 without chorioamnionitis and 988 with. Clinical and demographic data was analysed by Chi (categorical data) and t test (continuous data). Preterm deliveries were excluded in this study,  this is a viable strategy to try to get a slightly cleaner clinical data set.

Comment 1:  Due to the huge difference in the n of each group there is likely a Type 1 error in the t- test data. This is most apparent in table 2 Pregnancy outcomes where gestational age and birthweight are shown to be highly significantly different but they don't appear to be different at all.  The authors need to perform an Effect size test or Cohens test to prove that their analysis is not biased by the larger n of one group.

The authors do show a higher rate of mortality in infants of women with chorioamnionitis but in the long term analysis there is no significant findings in between the 2 groups  and the authors conclude that there is no long-term effect of in utero infection on infant and childhood morbidity.

Comment 2

Were the placentas form these deliveries assessed by a pathologist for grade of chorioamnionitis?  Would there be an association with outcome if the infection was confirmed to have crossed the placenta and amnion?

Author Response

Thank you to the authors for an interesting paper.  The paper is well written and a good read.

In this manuscript the authors investigate the hypothesis that term chorioamnionitis would be associated with a long term and higher infectious morbidity of the affected infants.

This is a retrospective study and had impressive numbers of patients/ data with an n of 330,610 without chorioamnionitis and 988 with. Clinical and demographic data was analysed by Chi (categorical data) and t test (continuous data). Preterm deliveries were excluded in this study,  this is a viable strategy to try to get a slightly cleaner clinical data set.

Comment 1:  Due to the huge difference in the n of each group there is likely a Type 1 error in the t- test data. This is most apparent in table 2 Pregnancy outcomes where gestational age and birthweight are shown to be highly significantly different but they don't appear to be different at all.  The authors need to perform an Effect size test or Cohens test to prove that their analysis is not biased by the larger n of one group.

Answer: We thank the reviewer for the important comment. In accordance, we performed a Cohen’s test for both gestations age and birthweight and found that the Point estimate for gestational age was -.108 [95%CI (-.171)- (-.046)] and .175 (95%CI .112-.237) for birthweight, proving a small effect size for both variables. Additionally, we added the following clarification to the results section: “To note, although gestational age and mean birthweight were found to have a statistically significant difference among the two study groups, this difference is clinically irrelevant as both outcomes reside well within the full term of gestational age with an appropriate birthweight.” (page 4, lines 154-157)

The authors do show a higher rate of mortality in infants of women with chorioamnionitis but in the long term analysis there is no significant findings in between the 2 groups  and the authors conclude that there is no long-term effect of in utero infection on infant and childhood morbidity.

Answer: We thank the reviewer for this comment and added a clarification in the discussion section: “Another limitation to note is the relatively high mortality rate (5.4%) among infants diagnosed with chorioamnionitis. This could suggest a possible selection bias towards more severe cases within this group. However, this study focused on offspring long-term infectious outcomes, and despite this bias, it's reassuring that, even among this severely affected population, we did not observe differences in future rates of infectious morbidity. (Page 8, lines 313-318).

Comment 2

Were the placentas form these deliveries assessed by a pathologist for grade of chorioamnionitis?  Would there be an association with outcome if the infection was confirmed to have crossed the placenta and amnion?

Answer: We thank the reviewer for the important notion regarding the placental pathology report. Unfortunately, this data was unavailable to us. Furthermore, in our institution (SUMC), placentas are sent for pathological assessment only in complicated pregnancies (preterm, abruption, chorioamnionitis) and not for all deliveries so this comparison is impossible between our study groups. However, it would be interesting to investigate placental pathologies only within the affected cohort of exposed children, but this investigation was not in the scope of our study. We added this limitation in the discussion section: "We acknowledge that we were unable to include the placental pathology report that would have confirmed the diagnosis in these deliveries, or further investigate the association between the histological grading of chorioamnionitis and the long-term infectious outcome of the offspring. Further studies are needed to evaluate this topic. (Page 8, Lines 329-332). 

We would like to thank the reviewer for the important comments that have certainly improved our manuscript.